# Characteristics of PM_2.5_ in an Industrial City of Northern China: Mass Concentrations, Chemical Composition, Source Apportionment, and Health Risk Assessment

**DOI:** 10.3390/ijerph19095443

**Published:** 2022-04-29

**Authors:** Wenyu Bai, Xueyan Zhao, Baohui Yin, Liyao Guo, Wenge Zhang, Xinhua Wang, Wen Yang

**Affiliations:** 1State Key Laboratory of Environmental Criteria and Risk Assessment, Chinese Research Academy of Environmental Sciences, Beijing 100012, China; bai.wenyu@craes.org.cn (W.B.); zhaoxy@craes.org.cn (X.Z.); yinbh@craes.org.cn (B.Y.); guo.liyao@craes.org.cn (L.G.); yangwen@craes.org.cn (W.Y.); 2College of Life and Environmental Sciences, Minzu University of China, Beijing 100081, China; 3National Institute of Metrology, Beijing 100029, China

**Keywords:** health risk, metal elements, traffic emissions, source apportionment

## Abstract

Urban and suburban PM_2.5_ samples were collected simultaneously during selected periods representing each season in 2019 in Zibo, China. Samples were analysed for water-soluble inorganic ions, carbon components, and elements. A chemical mass balance model and health risk assessment model were used to investigate the source contributions to PM_2.5_ and the human health risks posed by various pollution sources via the inhalation pathway. Almost 50% of the PM_2.5_ samples exceeded the secondary standard of China’s air quality concentration limit (75 µg/m^3^, 24 h). Water-soluble inorganic ions were the main component of PM_2.5_ in Zibo, accounting for 50 ± 8% and 56 ± 11% of PM_2.5_ at the urban and suburban sites, respectively. OC and OC/EC decreased significantly in the past few years due to enhanced energy restructuring. Pearson correlation analysis showed that traffic emissions were the main source of heavy metals. The Cr(VI) concentrations were 1.53 and 1.92 ng/m^3^ for urban and suburban sites, respectively, exceeding the national ambient air quality standards limit of 0.025 ng/m^3^. Secondary inorganic aerosols, traffic emissions, and secondary organic aerosols were the dominant contributors to PM_2.5_ in Zibo, with the total contributions from these three sources accounting for approximately 80% of PM_2.5_ and the remaining 20% attributed to traffic emissions. The non-carcinogenic risks from crustal dust for children were 2.23 and 1.15 in urban and suburban areas, respectively, exceeding the safe limit of 1.0 in both locations, as was the case for adults in urban areas. Meanwhile, the carcinogenic risks were all below the safe limit, with the non-carcinogenic and carcinogenic risks from traffic emissions being just below the limits. Strict control of precursor emissions, such as SO_2_, NOx, and VOCs, is a good way to reduce PM_2.5_ pollution resulting from secondary aerosols. Traffic control, limiting or preventing outdoor activities, and wearing masks during haze episodes may be also helpful in reducing PM_2.5_ pollution and its non-carcinogenic and carcinogenic health impacts in Zibo.

## 1. Introduction

Particulate matter with an aerodynamic diameter ≤ 2.5 μm, known as PM_2.5_, is suspended in the air and is composed of various chemicals and particle sizes. PM_2.5_ resides in the atmosphere for long periods and can penetrate deep into the lungs [1,2]. It has attracted significant attention due to its adverse effects on climate, visibility, and especially human health [3,4,5,6]. Long-term exposure to PM_2.5_ has been proven to increase the risk of lung cancer, dementia, cardiovascular disease, chronic respiratory diseases, and depressive-like responses [7,8,9,10,11,12].

Because PM_2.5_ poses human health risks, it is necessary to explore its toxicity. Carbonaceous species, water-soluble ionic species (WSI), and mineral dust are the major components of PM_2.5_ [13,14]. Several studies suggest that to reduce the health risks associated with PM_2.5_, chemical components and their sources should be considered, alongside mass concentrations [15,16,17]. Given that a substantial body of research highlights the toxicological importance of metals in PM_2.5_, an investigation into metal levels is necessary to understand their risks, and identification of their emission sources is needed to develop effective risk management plans [1].

Zibo is a highly industrialized cluster city of about 6000 km^2^ with a population of 4.7 million in the central area of Shandong Province. Its many heavy and light industries, such as ceramic and petrochemical industries, make it a highly polluted area [18,19,20,21,22]. Enrichment factor analysis, principal component analysis, and the PMF model were used to analyse the potential sources of PM_2.5_ and related metal elements in Zibo in recent years [19,23,24,25]. However, receptor models have not often been used in previous studies, and sampling stations have been concentrated in urban areas. Therefore, the main objectives of this work were to investigate (a) the characteristics of PM_2.5_ and its chemical composition (water-soluble ionic species, carbonaceous species, and element species) in urban and suburban areas, (b) the main source apportionment of PM_2.5_ using the CMB model, and (c) the source-specific health risks for adults and children induced by PM_2.5_-bound elements via inhalation.

## 2. Materials and Methods

### 2.1. Site Description and Sample Collection

The surrounding environments of the sampling sites, namely Nanding (urban area) and Lishan (suburban area), are shown in Figure 1. The urban site was located approximately 6 m above the ground on the second-storey rooftop of a local monitoring station (36°48′10.67″ N, 118°01′28.19″ E) and was surrounded by residential and administrative buildings. The suburban sampling site was approximately 12 m above the ground on the fourth-storey rooftop of a building (36°11′13.76″ N, 118°10′26.08″ E) and was surrounded by residential and commercial buildings.

A total of 45 PM_2.5_ samples were simultaneously collected at the two sites each day during the periods of 8–22 January (winter), 16–25 April (spring), 30 July–8 August (summer), and 16–25 October (autumn) in 2019. Each day’s sampling time was defined as running from 10:00 a.m. until 9:00 a.m. the next day (23 h). The samplers (ZR-3930; Zhongrui Inc., Qingdao, China) had two parallel channels and a flow rate of 16.7 L min^−1^ and used Teflon filters (diameter = 47 mm; Whatman Inc., Buckinghamshire, UK) and quartz filters (diameter = 47 mm; Pall, San Diego, CA, USA). The PM_2.5_ manual sampling method was in accordance with the latest national standard, HJ 656-2013 [26].

### 2.2. Gravimetric and Sample Analysis

Before sampling, quartz filters were baked in a muffle furnace at 550 °C for 3 h to remove carbonaceous pollutants. Before and after sampling, the filters were balanced under constant temperature (20 ± 1 °C) and humidity (50 ± 5%) for more than 24 h and weighed using an automatic filter-weighing system (AWS-1; Comde-Derenda GmbH, Germany, readability = 1 μg). The difference between two consecutive measurements was not more than 40 µg. After sampling, the filters were stored at −20 °C until analysis to prevent volatilization of volatile components [27]. The concentrations of PM_2.5_ were determined based on the values analysed from the Teflon filters.

Quartz filters were used to analyse the carbon contents and water-soluble inorganic ions (WSIIs). Organic carbon (OC) and elemental carbon (EC) were measured according to interagency monitoring of protected visual environments (IMPROVE) thermal optical reflectance (TOR; Model 2001A, Desert Research Institute, Reno, NV, USA). Because the oxidation temperatures of OC and EC are different, the IMPROVE temperature-programmed method was used to analyse the samples in this study. The heating program was divided into two stages: in the first stage, in the oxygen-free and pure-helium environment, the quartz filter was heated to 140 °C, 280 °C, 480 °C, and 580 °C to obtain the four components of OC (OC1, OC2, OC3, and OC4, respectively); in the second stage, in a helium atmosphere containing 2% oxygen, the temperature was gradually increased to 580 °C, 740 °C, and 840 °C to obtain the three components of EC (EC1, EC2, and EC3, respectively). Because OC forms optical pyrolyzed carbon (OPC) during the carbonization process, according to the IMPROVE analysis protocol, OC and EC were defined according to Equations (1) and (2), respectively [28].
OC = OC1 + OC2 + OC3 + OC4 + OPC (1)
EC = EC1 + EC2 + EC3 − OPC(2)

Water-soluble inorganic ions (Na^+^, NH_4_^+^, K^+^, Mg^2+^, Ca^2+^, Cl^−^, NO_3_^−^, and SO_4_^2−^) were measured using ion chromatography analysers (DIONEX ICS-1100 and ICS-2100, Thermo Company, Waltham, MA, USA). The samples were soaked in deionized water (10 mL), shaken well, leached under ultrasonic vibration for 15 min, and let stand for 5 min; then, the supernatant was injected for ion chromatographic analysis [21].

To analyse the concentrations of elements, one 1/2 of a Teflon filter was cut into fragments and solubilized in a 5 mL HF acid solution in a PTFE crucible. The solution was then progressively heated to 160 °C on an electric stove for 2 h and leached in 10 mL of 2% HCl. After cooling, the solution was moved into a plastic colorimetric tube before analysis of eleven trace elements (As, Cd, Co, Cr, Cu, Mn, Ni, Pb, Sr, V, and Zn) using inductively coupled plasma-mass spectrometry (ICP-MS, Agilent 7500a, Santa Clara, CA, USA). The other 1/2 of Teflon filter was put in a Nickel crucible and progressively heated to 300 °C to ash in a muffle furnace and kept in the furnace under constant temperature for about 20 min. After cooling, pure ethanol and 0.2 g NaOH were added to crucible and melted for 10 min. Following the extraction from boiling water, the solution was moved into a plastic tube with 1 mL HCl (12 mol/L). After dilution, the other trace elements (Al, Mg, Ca, Fe, and Ba) were analysed by an inductively coupled plasma-optical spectrometer (ICP-OES, Agilent, Santa Clara, CA, USA) [29].

### 2.3. Source Apportionment Model

Source apportionment was conducted with a CMB model. Considering that a large number of samples (usually >100) is required for statistical analysis and there some subjectivity and uncertainty are involved in the determination of the number of source types and the discrimination in PMF model [30,31], we chose a CMB model rather than a PMF model. The theory behind the CMB model has is introduced in detail in [31]. The fundamental relationship between the concentration at a receptor site and source information in the CMB model can be expressed as Equation (3):(3)$$C_{i}=\overset{p}{\underset{j=1}{\sum }}F_{ij}\times S_{j}\;$$
where *C**_i_* is the concentration (µg m^−3^) of elemental component *i* at a receptor site, *p* represents the number of sources, *F_ji_* is the fraction of the *i*th element from the *j*th source, and *S_j_* is the concentration of the contribution from the *j*th source to the receptor site.

In our study, 90 ambient PM_2.5_ samples and source profiles were input to CRAES CMB 1.0 to obtain the source apportionment results and contributions of various PM_2.5_-bound metals. CRAES CMB 1.0 is a Chinese-language version of the model derived from EPA CMB 8.2 by the Chinese Research Academy of Environmental Sciences, which includes an enumeration method [32]. Based on the emission inventory and industrial structure of Zibo, we chose the potential sources and conducted source sampling and analysis work. The squared regression coefficient (R^2^), sum of residual squares value (*Χ*^2^), degrees of freedom (DF), and percentage of explained mass to sample total mass were used to evaluate the validity of source contribution measures [33]. Table 1 summarizes the performance measures of the CMB model used in the present study.

### 2.4. Health Risk Assessment Model

The investigated model was introduced by the United States Environmental Protection Agency to quantify health risks (carcinogenic and non-carcinogenic) and human exposure to elements in dust [34]. The three main exposure pathways in humans are direct ingestion, inhalation, and dermal contact. PM_2.5_-bound metals cause human health hazards mainly via the respiratory system [31], so we only considered the inhalation exposure pathway in this study. Exposure is expressed in terms of a daily dose (mg kg^−1^ day^−1^), and the doses received through inhalation were calculated with Equations (4) and (5) [35,36]. Ba, Co, Cr(VI), and Mn exhibited non-carcinogenic risks, whereas As, Cd, Co, Cr(VI), and Ni exhibited carcinogenic risks in this study [37,38]. Because the toxicity of Cr is attributed to its hexavalent state, its concentrations were scaled by a factor of 1/7, given that atmospheric Cr(VI) is reported to have a ratio of 1/6 relative to Cr(III) [39,40].
(4)$${\mathrm{ADD}}_{\mathrm{inh}}=C\times \frac{\mathrm{InhR}\times \mathrm{EF}\times \mathrm{ED}}{\mathrm{PEF}\times \mathrm{BW}\times \mathrm{AT}}$$
(5)$$\mathrm{LADD}=\frac{C\times \mathrm{EF}}{\mathrm{AT}\times \mathrm{PEF}}\times \left(\frac{{\mathrm{InhR}}_{\mathrm{child}}\times {\mathrm{ED}}_{\mathrm{child}}}{{\mathrm{BW}}_{\mathrm{child}}}+\frac{{\mathrm{InhR}}_{\mathrm{adult}}\times {\mathrm{ED}}_{\mathrm{adult}}}{{\mathrm{BW}}_{\mathrm{adult}}}\right)$$
where ADD_inh_ (mg kg^−1^ day^−1^) is the dose obtained via inhalation, and LADD (mg kg^−1^ day^−1^) is the lifetime average daily dose of carcinogens. The exposure parameters are listed in Table 2.

To assess the health risks due to trace element exposure in PM_2.5_, the hazard quotient (HQ), hazard index (HI), and carcinogenic risk (RI) techniques were employed and calculated with Equations (6)–(8) [35,36].
(6)HQ=ADD×RfD
(7)$$\mathrm{HI}=\sum^{\;}{\mathrm{HQ}}_{i}$$
(8)RI=LADD×SF
where RfD (mg kg^−1^ day^−1^) is the corresponding reference dose, and SF is the corresponding slope factor. The values of RfD and SF used in this study are listed in Table 3.

## 3. Results and Discussion

### 3.1. PM_2.5_ Mass Concentrations

Figure 2 shows the PM_2.5_ mass concentrations in winter, spring, summer, and autumn during the sampling period at the urban and suburban sites in Zibo. The minimum, median, and maximum concentrations of urban PM_2.5_ were 20, 56, and 313 µg/m^3^, respectively, with 48.89% of the values exceeding the secondary standard of China’s air quality concentration limit (75 µg/m^3^, 24 h) [44]. The suburban minimum, median, and maximum concentrations of PM_2.5_ were 20, 65, and 212 µg/m^3^, respectively, with 42.22% of data representing concentrations > 75 µg/m^3^. Furthermore, 80% of the values at both sites exceeded the primary standard limit (35 µg/m^3^, 24 h), and 100% exceeded the WHO air quality guidelines for PM_2.5_ (15 µg/m^3^, 24 h) [44,45]. By comparing the statistics of the two sites, we can see that the median and maximum concentrations at the urban site were greater than those at the suburban site, especially the maximum values. Compared with sites in the North China Plain (NCP; northwest of Zibo), the proportion of polluted weather (PM_2.5_ > 75 µg/m^3^) at the urban (67%) and suburban (67%) sites during the winter sampling period were greater than that at the NCP sites (50%). However, the overall concentrations in summer were <75 µg/m^3^ at both the urban and suburban sites, whereas the NCP sites experienced haze pollution during their intensive summer observation period [46]. In other words, Zibo experienced greater winter pollution than the NCP but less summer pollution.

### 3.2. Chemical Composition Levels

#### 3.2.1. WSII Levels

The annual mean WSII concentrations of PM_2.5_ at the urban and suburban sites (Table 4) were 42.2 ± 35.0 and 44.0 ± 31.3 µg/m^3^, accounting for 50 ± 8% and 56 ± 11% of the total PM_2.5_, respectively, which is close to the results for 2018 [27]. This result shows that WSIIs were the main components of PM_2.5_ in both urban and suburban areas. The secondary water-soluble inorganic ions composed of sulphate (SO_4_^2−^), nitrate (NO_3_^−^), and ammonium (NH_4_^+^) aerosol (SNAs) were the major contributors to WSIIs, and their annual average concentrations were 37.8 ± 32.7 µg/m^3^ (approximately 44% of the PM_2.5_ in urban site) and 39.7 ± 29.8 µg/m^3^ (approximately 50% of the PM_2.5_ in suburban site). SNAs are formed in the atmosphere from gaseous precursors, for example, sulphur dioxide (SO_2_), nitrogen oxides (NOx), and ammonia (NH_3_), through complex chemical reactions [47]. The ratio of SNA/PM_2.5_ was highest in autumn (56% for both urban and suburban sites), which is indicative of intense chemical reactions in this season.

The ratio of NO_3_^−^ to SO_4_^2−^ ([NO_3_^−^/SO_4_^2−^]) has been used as an indicator of the relative importance of mobile vs. stationary sources of nitrogen and sulphur in the atmosphere [48,49,50]. Ratios > 1.0 indicate that the particle sources at the sampling site are likely dominated by mobile sources, whereas stationary sources predominate at ratios < 1.0. In this study, the ratios of NO_3_^−^/SO_4_^2−^ were 2.5, 2.2, 0.5, and 3.9 (2.5, 2.7, 0.8 and 4.2) in winter, spring, summer, and autumn, respectively, with an annual average ratio of 2.3 (2.5) at the urban site (suburban site, Table 4). Note that, in summer, a lower NO_3_^−^/SO_4_^2−^ ratio may be ascribed to high temperatures, which lead to the evaporation of NH_4_NO_3_, whereas high ratios in other seasons are more likely related to traffic emissions from roads near the sampling sites [51].

There was a typical haze episode from 10 to 14 January 2019 in Zibo according to the definition of haze episode [27]; Figure 3 gives the daily concentrations of PM_2.5_, SNA, and NO_3_^−^/SO_4_^2−^ during this episode. The average concentrations of PM_2.5_ were 197 µg/m^3^ and 180 µg/m^3^ at the urban and suburban sites, respectively, and the peak values were 312 and 212 µg/m^3^, respectively. Obviously, SNAs were the main components in PM_2.5_ during this episode, and the peak values of SNA/PM_2.5_ were over 60%. SNA components ranked in the order of NO_3_^−^ > SO_4_^2−^ > NH_4_^+^, which was different from Beijing, where the SO_4_^2−^ ranked first [50]. In addition, NO_3_^−^/SO_4_^2−^ was above 1.0 and increased during the accumulation period. The above information demonstrates that NO_3_^−^ played a predominant role in the formation of heavy haze, and mobile sources had a relatively important contribution.

#### 3.2.2. OC and EC Levels

OC and EC are also major components of PM_2.5_, and their daily concentrations are shown in Figure 4. The annual concentrations of OC in PM_2.5_ in urban and suburban areas were 10.9 and 11.2 µg/m^3^, respectively, which are lower than those in other Chinese cities, e.g., 13.8 µg/m^3^ in Changzhou [51] and 14.4 µg/m^3^ in Wuhan [52], and are comparable to those measured in Guangzhou (9.4 µg/m^3^) [5]. The annual EC concentrations were 4.7 µg/m^3^ in urban and 5.1 µg/m^3^ in suburban areas, which are close to those of Changzhou (5.4 µg/m^3^) [51], Wuhan (5.2 µg/m^3^), and Guangzhou (6.2 µg/m^3^) [5]. Obvious similar seasonal variations in OC and EC were observed in both urban and suburban areas, with the highest average concentrations occurring in winter: 18.6 and 6.8 µg/m^3^ in urban and 19.1 and 8.2 µg/m^3^ in suburban areas, respectively. Moreover, the ratios of OC/PM_2.5_ were highest in winter, accounting for 19% and 20% of the PM_2.5_ in urban and suburban areas, respectively. The lowest concentrations of OC and EC were observed in summer.

The OC/EC ratio was used to indicate the existence of secondary organic carbon (SOC). More than 60% of OC/EC ratios in this study exceeded 2.0, which reflects the fact that SOC is a significant component of PM_2.5_ [53]. However, the annual average OC/EC ratio in this study was only 2.4, which is much lower than that reported in a previous study (OC/EC = 6.68, OC = 31.97 ± 20.43 μg/m^3^, EC = 5.01 ± 2.68 μg/m^3^; sampling time = 2006–2007) [25]. Clearly, the distinction between annual average OC concentrations caused this result. This might be attributed to the partial replacement of coal-fired energy with clean energy over the past decade, especially after the Chinese State Council implemented the Air Pollution Prevention and Control Action Plan in September 2013. According to a study by Li et al., study [54], OC emissions decreased by 31.7% in Beijing and surrounding areas (including Zibo) during the period of 2014–2017 as a result of strict governance of residential fuel and enhanced energy restructuring by using cleaner natural gas and electricity replacing coal.

#### 3.2.3. Element Levels

The seasonal average concentrations of various elements in PM_2.5_ are shown in Figure 5. The total elemental concentrations accounted for 1.96% and 1.33% of the mean PM_2.5_ concentrations at the urban and suburban site, respectively. In urban PM_2.5_ samples, Ca exhibited the highest levels (453.9), followed by Al (384.7), Zn (234.6), Fe (192.0), Mg (127.5), Mn (34.5), Pb (34.0), Ba (12.4), Cu (11.4), Cr (10.7), As (6.0), Ni (4.1), Sr (3.6), V (1.7), Cd (1.6), and Co (0.3; median values, ng/m^3^). Similarly, Ca exhibited the highest levels in the suburban PM_2.5_ samples (261.7), followed by Al (152.0), Fe (116.4), Zn (106.7), Mg (100.1), Mn (24.8), Pb (18.4), Cr (13.5), Cu (5.3), As (3.8), Ni (3.0), Ba (2.6), V (1.2), Sr (1.1), Cd (0.7), and Co (0.2; medians, ng/m^3^). The annual median concentrations of urban and suburban heavy metals Pb, Cd, and As were lower than the national ambient air quality standards limits of 500 ng/m^3^, 5 ng/m^3^, and 6 ng/m^3^, respectively. However, the Cr(VI) concentrations exceeded the limit of 0.025 ng/m^3^ [45]. Shandong province, where the Zibo is located, has the highest Cr emissions among all provinces in China due to its high capacity for steel production and coal consumption of industrial boilers [55].

Figure 6 shows the annual trends of 16 metal elements in urban and suburban areas, and their Pearson correlations are shown in Figure 7. The urban and suburban sites have several common characteristics. Al, Ca, and Fe were significantly correlated with one another (*p* < 0.01, *r* > 0.48), and their concentrations increased significantly in spring, which is the season of frequent sand/dust storm weather in northern China [14], increasing these crustal elements. Additional, Mn, Co, Cu, Zn, Cd, and Pb were also significantly correlated with one another (*p* < 0.01, *r* > 0.86), all of which are typically elements of motor vehicle emissions sources or classic traffic-source-related elements [56,57]; most showed a significant increase in winter and autumn. The probable reasons are that vehicle emissions have strong time sensitivity, being highest in winter and lowest in summer [58], with clean travel modes, such as electric bicycles, being more commonly used in summer. Arsenic, a representative coal-combustion element, showed higher concentrations in winter than in other seasons due to increased demands for space heating [13].

However, there were some distinctions between the urban and suburban sites; e.g., V and Ni were not correlated with any of the other 15 elements at the suburban site, whereas the concentration of Ni was significantly lower in winter. This is probably due to the fact that agricultural and construction activities are more common in suburban areas than in urban areas and are less prevalent in winter, resulting in a significant decrease in the use of related off-road machinery and corresponding exhaust emissions [59,60]. Sr and Ba, which are typical alkaline-earth metals emitted by industrial processes such as alloy smelting, ceramic production, etc., were significantly related to each other in urban samples (*p* < 0.01, *r* = 0.81).

### 3.3. Source Apportionment Using CMB

#### 3.3.1. CMB Source Profiles

Chemical mass balance models for source apportionment require information about the chemical characteristics of the sources that are likely to affect pollutant concentrations at receptor sites [33]. The weight percentages of the PM_2.5_-bound chemical species in all sources (crustal dust, coal boilers, graphite production, aluminium smelting, glass manufacturing, iron smelting, traffic emissions, secondary inorganic aerosols, and secondary organic aerosols) are presented in Figure 8.

It is evident from the source profiles that the major components of crustal dust are Ca, OC, SO_4_^2−^, Si, and Al (Ca, Si, and Al are crustal elements). The high percentage of OC is an indication of the mixing of dust and organic matter during aging or by entrainment of organic materials from soils [61]. The coal boiler profile is dominated by SO_4_^2−^, OC, NH_4_^+^, Na, Cl^−^, Si, and Al, with abundances of 23.9%, 18.0%, 9.9%, 3.4%, 2.5%, 2.4%, and 2.0%, respectively. This is consistent with the results of a study by Han et al. [62] in which OC, EC, SO_4_^2−^, Ca, Si, Fe, and Al were found to be the abundant species in coal combustion fly ash from Xining City, China. The source profiles for graphite production and glass manufacturing are similar; therefore, SO_4_^2−^, NH_4_^+^, and OC were the major components, and Zn [63] and Na [64] were also enriched in graphite production and glass manufacturing, respectively. Aluminium smelting and iron smelting are characterized by large amounts of Al (19.6%) and Fe (9.6%) [1], respectively. Carbon components are the most abundant species in traffic emissions. It is worth noting that the abundance of EC in traffic emissions is 14.3%, which is 5–27 times that in other profiles. EC was significantly correlated with traffic-source-related elements (Mn, Co, Cu, Zn, Cd, and Pb) (*p* < 0.01) in ambient PM_2.5_ samples. Hence, EC can be used as a key marker to distinguish traffic emissions from other sources, as it primarily arises from engines [13].

#### 3.3.2. CMB Source Apportionment

The CMB results identified nine sources of PM_2.5_ during the winter, spring, summer, and autumn sampling periods, including crustal dust (CD), coal boilers (CB), graphite production (GP), aluminium smelting (AS), glass manufacturing (GM), iron smelting (IS), traffic emissions (TE), secondary inorganic aerosols (SIA, including NO_3_^−^, SO_4_^2−^, and NH_4_^+^), and secondary organic aerosols (SOA; Table 5). The top annual contributors were ranked as SIA > TE > SOA, with percentage contributions of 43.34%, 24.03%, and 9.78% in urban and 49.44%, 20.49%, and 10.57% in suburban areas, respectively. The percentage contributions of SIA were nearly 50%, which is consistent with the high concentrations of SNAs in PM_2.5_. The high contribution of traffic emissions might be attributable to the rapid increase in motor vehicle numbers over the past 10 years (from 443,179 to 1,136,377; Zibo Statistical Yearbook, 2010, 2019; http://tj.zibo.gov.cn/col/col886/index.html; accessed on 31 October 2021). Apart from TE and SIA, SOAs also played an important role as a PM_2.5_ source in Zibo. Zibo is an important petrochemical industry base of China, and there were amounts of volatile organic compounds (VOCs) emissions (100 kt/year) [18] that could result in the formation of SOAs [25,65]. In addition, the percentage contribution of crustal dust was lower than that in in Tianjin (26.4%), Langfang (21.4%), Baoding (16.9%) [66], and Zibo in 2006 (29.2%) [25]. Combined with the low percentage of elements, the main reason should be that the Zibo government implemented many strict dust management policies in the last few years [67]. Although the percentage contributions of coal boilers, graphite production, aluminium smelting, glass manufacturing, and iron smelting were all lower than 3%, their contributions to gaseous precursors, such as SO_2_ and NOx, could not be ignored because of the high contribution of SIA. The contribution rates were similar between the urban and suburban sites.

### 3.4. Risk Characterization

Because using the total concentration of heavy metals not only overestimated the human health risk but also inaccurately identified the high-risk pollution sources [68], we chose a source-specific human health risk assessment model based on the inhalation mode of exposure. The results are shown in Table 6.

Crustal dust had the highest non-carcinogenic risk (1.26 for adults and 2.23 for children at the urban site vs. 0.65 for adults and 1.15 for children at the suburban site), and except for 0.65, the other three values exceeded the safe limit 1.00. This was caused by the high HQ values of Mn (Table 6). Meanwhile, Mn had the highest HQ values of all sources, which is a similar result to that reported in a previous study [34]. Furthermore, the high HI values for children from crustal dust indicate that children are more vulnerable to non-carcinogenic health risks [34]. For the other four sources, both the HQ and accumulative HI values of individual emission sources identified in this study were lower than the safe limit of 1.00 for both adults and children, which indicates that there were no adverse non-carcinogenic health risks. However, the HI values of traffic emissions were 0.78 and 0.86 for children in urban and suburban areas, respectively, which are only a little lower than 1. For cancer risk (RI), the model estimated a non-significant cancer risk for both adults and children, with RI values lower than 1.00 × 10^−6^, suggesting that the carcinogenic risk from these elements is negligible. In addition, the RI values were highest for traffic emissions at both the urban and LS sit (1.60 × 10^−^^7^ and 1.33 × 10^−7^, respectively).

## 4. Conclusions

In this study, we investigated the characteristics of PM_2.5_ chemical composition, source apportionment, and health risk in Zibo. The results show that (1) almost 50% of PM_2.5_ samples exceeded the secondary standard of China’s air quality concentration limit (75 µg/m^3^, 24 h). Water-soluble inorganic ions were the main component of PM_2.5_ in Zibo, accounting for 50 ± 8% and 56 ± 11% at the urban and suburban site, respectively. Except in summer, the ratios of NO_3_^−^ to SO_4_^2−^ exceeded 1, implying important contributions from mobile sources. The annual average concentrations of OC accounted for 15% of PM_2.5_, and enhanced energy restructuring resulted in a decrease in OC and OC/EC in the past few years. Ni exhibits significant seasonal variation due to agricultural and construction activities in suburban areas. The total elemental concentrations accounted for 1.96% and 1.33% of the mean PM_2.5_ concentrations at the urban and suburban sites, respectively, and the Cr(VI) concentrations were 1.53 and 1.92 ng/m^3^, respectively, which exceed the national ambient air quality Standards limit of 0.025 ng/m^3^ due to its large Zibo’s high capacity for coal consumption of industrial boilers. (2) The source apportionment results show that secondary inorganic aerosols, traffic emissions, and secondary organic aerosols were the dominant contributors to PM_2.5_ in Zibo. The total contributions of the three sources accounted for approximately 80% of PM_2.5_. Strict control of precursor emissions, such as SO_2_, NOx, and VOCs, is a good way to reduce the PM_2.5_ pollution resulting from secondary aerosols. Traffic emissions contributed more than 20% to PM_2.5_ and were the main source of heavy metals, according to combined Pearson correlation analysis of elements, so proper control measures should be taken. (3) The non-carcinogenic risks from crustal dust were 2.23 and 1.15 in urban and suburban areas, respectively, exceeding the safe limit for children and the limit for adults at the urban site. Meanwhile, the carcinogenic risks were all below the safe limit. However, non-carcinogenic and carcinogenic risks from traffic emissions were just below the safe limit. These results may help local authorities to design strategies for reducing PM_2.5_ pollution and limiting health impacts in Zibo during haze episodes; for example, by traffic control, by limiting or preventing outdoor activities, and by wearing masks.

## Figures and Tables

**Figure 1 ijerph-19-05443-f001:**
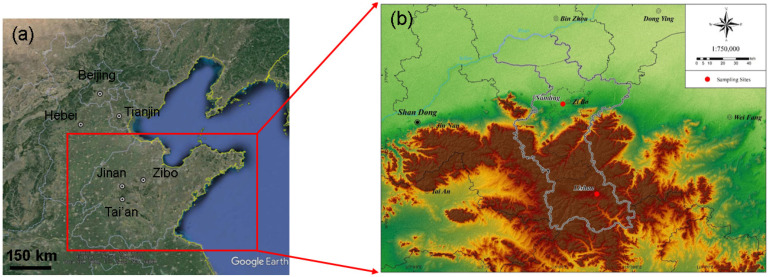
Geographical information of: (**a**) North China Plain, and (**b**) sampling sites of Nanding (urban site) and Lishan (suburban site) in Zibo.

**Figure 2 ijerph-19-05443-f002:**
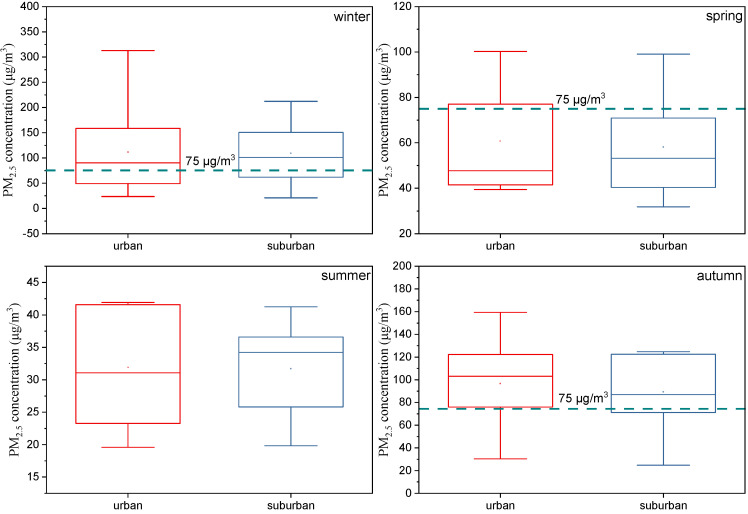
Sampled concentrations of PM_2.5_ in urban (red) and suburban areas (blue) during each season. Each subfigure shows the daily mean values (dots), median (horizontal line), central 50% of data (25–75th percentile; box), and the central 90% of data (5–95th percentile; whiskers).

**Figure 3 ijerph-19-05443-f003:**
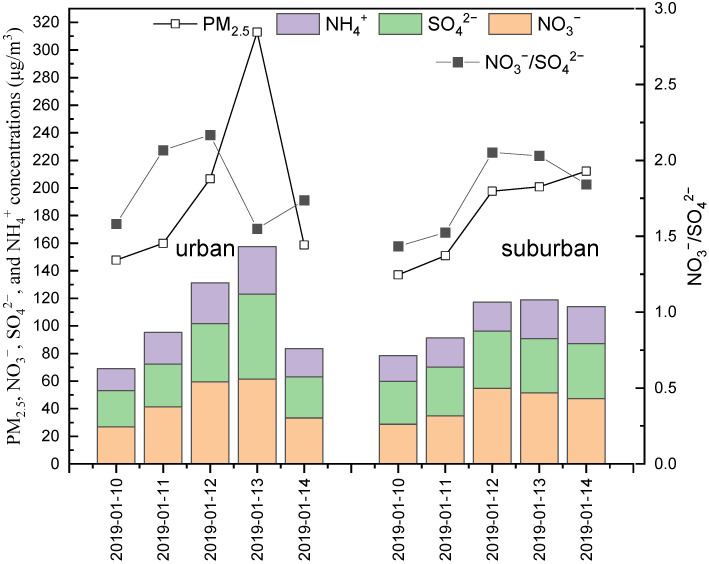
Variation of SNA during haze episode (**left**: urban; **right**: suburban).

**Figure 4 ijerph-19-05443-f004:**
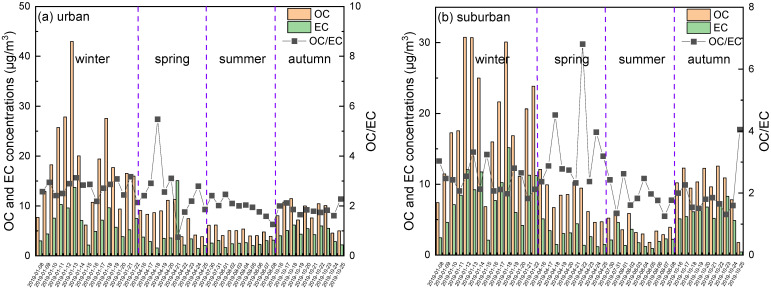
Concentrations of OC and EC and OC/EC ratio in urban (*n* = 45) and suburban (*n* = 45) areas during the sampling period.

**Figure 5 ijerph-19-05443-f005:**
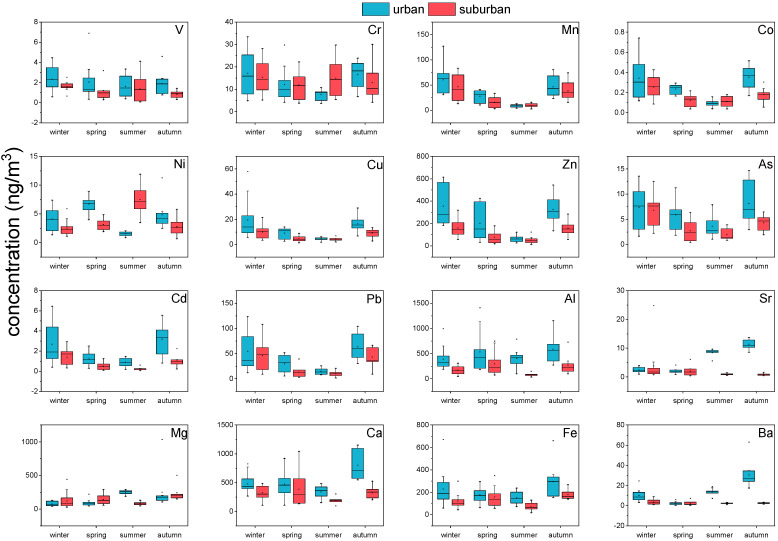
Boxplots showing the descriptive statistics of the sampled concentrations of PM_2.5_-metals. Each subfigure shows the mean (dot), median (horizontal line), central 50% of data (25–75th percentile; box), and central 90% of data (5–95th percentile; whiskers).

**Figure 6 ijerph-19-05443-f006:**
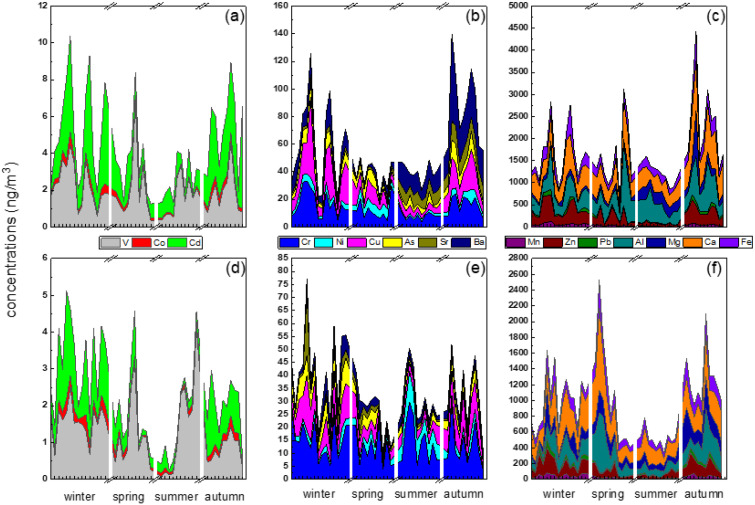
Daily variation series of PM_2.5_ metals (urban (**a**–**c**); suburban (**d**–**f**)).

**Figure 7 ijerph-19-05443-f007:**
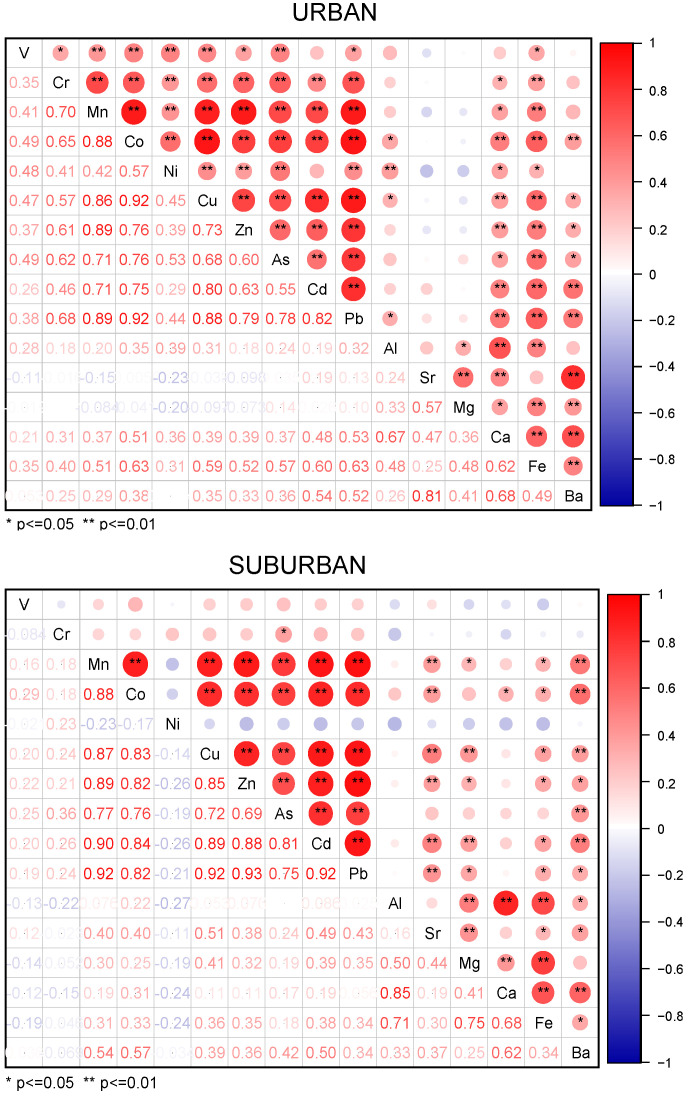
Pearson correlation of elements in urban (*n* = 45) and suburban (*n* = 45) samples.

**Figure 8 ijerph-19-05443-f008:**
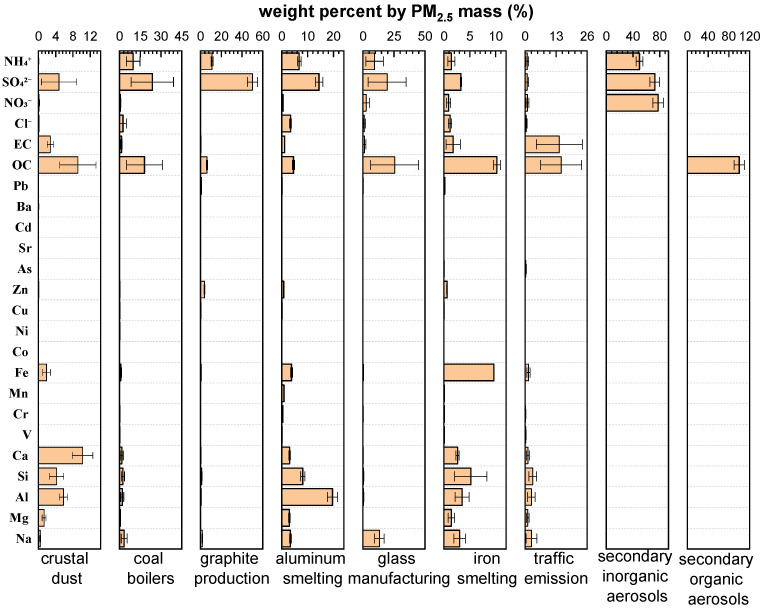
PM_2.5_ source profiles (weight percent by mass) in this study.

**Table 1 ijerph-19-05443-t001:** Chemical mass balance model performance test results during the study period.

Sampling Duration	Sampling Site	R^2^	CHI^2^	DF	Percentage of Explained Mass to Sample Total Mass
Winter	urban (*n* = 15)	0.94	0.39	5	83.9
	suburban (*n* = 15)	0.89	0.82	5	83.0
Spring	urban (*n* = 10)	0.88	1.06	8	82.7
	suburban (*n* = 10)	0.93	0.69	7	88.4
Summer	urban (*n* = 10)	0.94	1.57	5	84.1
	suburban (*n* = 10)	0.83	1.48	8	80.7
Autumn	urban (*n* = 10)	0.98	0.24	5	82.3
	suburban (*n* = 10)	0.94	0.65	5	84.4
Annual	urban (*n* = 10)	0.93	0.34	5	82.8
	suburban (*n* = 10)	0.93	0.39	5	87.3

Note: The criteria for acceptable CMB results included the square regression coefficient, R^2^ ≥ 0.8; the sum of residual square value, CHI^2^ ≤ 4; the degree of the freedom, DF ≥ 5; and the percentage of explained mass to sample total mass, ranging from 80 to 120%.

**Table 2 ijerph-19-05443-t002:** Values of exposure parameters for children and adults.

Parameter	Definition	Unit	Value		Reference
			Children	Adult	
*C*	Exposure-point concentration	mg kg^−1^	Present study ^a^		
InhR	Inhalation rate	m^3^ day^−1^	9.0	16.1	Refs. [41,42]
EF	Exposure frequency	day year^−1^	350	350	Ref. [34]
ED	Exposure duration	year	6	24	Ref. [34]
PEF	Particle emission factor	m^3^ kg^−1^	1.36 × 10^9^	1.36 × 10^9^	Ref. [34]
BW	Average body weight	kg	20.5	65.0	Refs. [41,42]
AT	Averaging time	day	365 × ED ^b^ 365 × 78 ^c^	365 × ED ^b^ 365 × 78 ^c^	Ref. [43]

Note: ^a^ By using the concentration of the individual metal (profile) of each source for *C*. ^b^ For non-carcinogenic risk assessment. ^c^ For carcinogenic risk assessment.

**Table 3 ijerph-19-05443-t003:** Values of RfD and SF used in the present study [37,43].

Element	RfD	SF
As		1.51 × 10^1^
Ba	1.43 × 10^−4^	
Cd		6.30
Co	5.71 × 10^−6^	9.80
Cr	2.86 × 10^−5^	4.20 × 10^1^
Mn	1.43 × 10^−5^	
Ni		8.40 × 10^−1^

**Table 4 ijerph-19-05443-t004:** Seasonal average WSII concentrations and PM_2.5_ ratios in urban and suburban areas.

	Winter *n* = 15		Spring *n* = 10		Summer *n* = 10		Autumn *n* = 10		Annual *n* = 45	
Urban	Mean	SD	Mean	SD	Mean	SD	Mean	SD	Mean	SD
PM_2.5_ (μg/m^3^)	112	±75	61	±23	32	±8	97	±39	79	±58
WSIIs (μg/m^3^)	59.1	±46.8	31.6	±14.1	14.8	±4.9	54.6	±23.3	42.2	±35.0
Cl^−^ (μg/m^3^)	4.5	±2.4	1.4	±1.2	0.1	±0.1	1.8	±0.9	2.2	±2.3
**Ratios**										
OC/EC	2.7	±0.3	2.6	±1.2	2.0	±0.3	1.9	±0.2	2.3	±0.7
NO_3_^−^/SO_4_^2−^	2.5	±0.9	2.2	±1.2	0.5	±0.3	3.9	±1.6	2.3	±1.6
WSIIs/PM_2.5_ (%)	49	±10	51	±7	46	±5	56	±5	50	±8
SNA/PM_2.5_ (%)	42	±11	46	±6	39	±6	51	±8	44	±9
OC/PM_2.5_ (%)	19	±4	13	±5	16	±4	10	±2	15	±5
EC/PM_2.5_ (%)	7	±2	6	±3	8	±2	5	±1	7	±2
	**Winter *n* = 15**		**Spring *n* = 10**		**Summer *n* = 10**		**Autumn *n* = 10**		**Annual *n* = 45**	
**Suburban**	**Mean**	**SD**	**Mean**	**SD**	**Mean**	**SD**	**Mean**	**SD**	**Mean**	**SD**
PM_2.5_ (μg/m^3^)	109	±60	58	±22	32	±7	89	±30	76	±49
WSIIs (μg/m^3^)	59.2	±39.3	37.2	±19.3	17.3	±4.0	54.7	±22.1	44.0	±31.3
Cl^−^ (μg/m^3^)	3.3	±1.2	1.8	±2.1	0.4	±0.2	1.3	±0.7	1.9	±1.7
**Ratios**										
OC/EC	2.5	±0.5	3.4	±1.3	1.9	±0.4	2.0	±0.7	2.4	±1.0
NO_3_^−^/SO_4_^2−^	2.5	±0.8	2.7	±1.3	0.8	±0.4	4.2	±1.7	2.5	±1.6
WSIIs/PM_2.5_ (%)	52	±9	62	±16	54	±7	60	±6	56	±11
SNA/PM_2.5_ (%)	46	±11	53	±10	48	±8	56	±7	50	±10
OC/PM_2.5_ (%)	20	±6	15	±3	13	±4	11	±3	15	±6
EC/PM_2.5_ (%)	8	±3	5	±1	7	±2	6	±2	7	±3

**Table 5 ijerph-19-05443-t005:** Percentage contributions of each source type to PM_2.5_ concentrations (%) in each season.

Sampling Duration	Sampling Site	Percentage Contribution of Each Source Type									
		CD	CB	GP	AS	GM	IS	TE	SIA	SOA	Others
8–22 January (winter)	urban	6.4	2.9	2.7	1.7	1.5	1.61	26.0	43.8	12.6	0.8
	suburban	6.1	2.6	1.3	1.1	1.2	1.4	22.5	45.4	13.4	5.0
16–25 April (spring)	urban	13.6	2.1	2.2	0.8	1.2	1.9	18.8	42.2	8.2	9.0
	suburban	7.2	1.5	2.3	0.3	0.6	0.9	14.5	52.5	10.9	9.4
30 July–8 August (summer)	urban	10.1	1.8	1.8	1.6	1.8	2.2	27.8	34.8	9.9	8.2
	suburban	6.6	1.2	1.1	1.1	1.0	1.3	25.7	35.8	6.8	19.4
16–25 October (autumn)	urban	5.8	1.9	1.9	1.3	1.5	1.5	24.4	46.3	5.5	9.9
	suburban	4.6	1.9	2.1	0.4	1.0	1.4	23.1	51.4	7.3	6.7
Annual	urban	7.8	2.3	2.2	1.4	1.8	0.9	24.0	43.3	9.8	6.4
	suburban	5.6	1.6	1.9	0.7	1.0	1.2	20.5	49.4	10.6	7.5

**Table 6 ijerph-19-05443-t006:** Source-specific non-carcinogenic and carcinogenic risks of heavy metals in ambient PM_2.5_ collected from urban and suburban areas in Zibo.

Non-Carcinogenic Risk: Hazard Quotient (HQ) of Each Heavy Metal and Hazard Index (HI)										
	Adults					Children				
	CD	CB	GP	GM	TE	CD	CB	GP	GM	TE
**Urban site**										
HQ_Cr_	1.72 × 10^−3^	2.74 × 10^−3^	4.46 × 10^−3^	0.00	4.38 × 10^−2^	3.05 × 10^−3^	4.85 × 10^−3^	7.90 × 10^−3^	0.00	7.77 × 10^−2^
HQ_Mn_	**1.20**	2.18 × 10^−1^	2.83 × 10^−1^	0.00	3.70 × 10^−1^	**2.12**	3.86 × 10^−1^	5.02 × 10^−1^	0.00	6.57 × 10^−1^
HQ_Co_	7.36 × 10^−3^	6.55 × 10^−3^	7.97 × 10^−3^	0.00	5.12 × 10^−3^	1.30 × 10^−2^	1.16 × 10^−2^	1.41 × 10^−2^	0.00	9.08 × 10^−3^
HQ_Ba_	4.94 × 10^−2^	1.31 × 10^−3^	3.92 × 10^−3^	0.00	1.92 × 10^−2^	8.76 × 10^−2^	2.32 × 10^−3^	6.95 × 10^−3^	0.00	3.40 × 10^−2^
HI	**1.26**	2.28 × 10^−1^	3.00 × 10^−1^	0.00	4.39 × 10^−1^	**2.23**	4.05 × 10^−1^	5.31 × 10^−1^	0.00	7.77 × 10^−1^
**Suburban site**										
HQ_Cr_	8.28 × 10^−4^	2.99 × 10^−3^	3.06 × 10^−3^	3.33 × 10^−3^	4.54 × 10^−2^	1.47 × 10^−3^	5.31 × 10^−3^	5.42 × 10^−3^	5.91 × 10^−3^	8.05 × 10^−2^
HQ_Mn_	6.36 × 10^−1^	2.70 × 10^−1^	2.22 × 10^−1^	9.53 × 10^−2^	4.29 × 10^−1^	**1.13**	4.79 × 10^−1^	3.94 × 10^−1^	1.69 × 10^−1^	7.60 × 10^−1^
HQ_Co_	2.17 × 10^−3^	4.37 × 10^−3^	3.35 × 10^−3^	8.69 × 10^−4^	3.26 × 10^−3^	3.85 × 10^−3^	7.75 × 10^−3^	5.95 × 10^−3^	1.54 × 10^−3^	5.78 × 10^−3^
HQ_Ba_	8.04 × 10^−3^	4.42 × 10^−4^	8.97 × 10^−4^	1.47 × 10^−4^	6.72 × 10^−3^	1.43 × 10^−2^	7.83 × 10^−4^	1.59 × 10^−3^	2.61 × 10^−4^	1.19 × 10^−2^
HI	6.47 × 10^−1^	2.78 × 10^−1^	2.30 × 10^−1^	9.97 × 10^−2^	4.84 × 10^−1^	**1.15**	4.93 × 10^−1^	4.07 × 10^−1^	1.77 × 10^−1^	8.59 × 10^−1^
**Carcinogenic Risk: Carcinogenic Risk (RI_i_) of Each Heavy Metal and Total Carcinogenic Risk (RI)**										
	**Urban Site**					**Suburban Site**				
	**CD**	**CB**	**GP**	**GM**	**TE**	**CD**	**CB**	**GP**	**GM**	**TE**
RI_Cr_	2.62 × 10^−9^	4.17 × 10^−9^	6.79 × 10^−9^	0.00	6.68 × 10^−8^	1.26 × 10^−9^	4.56 × 10^−9^	4.66 × 10^−9^	5.08 × 10^−9^	6.92 × 10^−8^
RI_Co_	5.22 × 10^−10^	4.65 × 10^−10^	5.66 × 10^−10^	0.00	3.64 × 10^−10^	1.54 × 10^−10^	3.10 × 10^−10^	2.38 × 10^−10^	6.17 × 10^−11^	2.31 × 10^−10^
RI_Ni_	3.18 × 10^−10^	1.29 × 10^−9^	1.35 × 10^−9^	0.00	4.98 × 10^−10^	1.50 × 10^−10^	1.37 × 10^−9^	9.04 × 10^−10^	3.14 × 10^−10^	5.05 × 10^−10^
RI_As_	2.66 × 10^−10^	5.46 × 10^−10^	7.23 × 10^−10^	0.00	9.15 × 10^−8^	8.50 × 10^−11^	3.91 × 10^−10^	3.29 × 10^−10^	6.06 × 10^−10^	6.30 × 10^−8^
RI_Cd_	3.45 × 10^−10^	2.30 × 10^−10^	1.08 × 10^−8^	0.00	9.21 × 10^−10^	1.11 × 10^−10^	1.34 × 10^−10^	4.08 × 10^−9^	3.12 × 10^−10^	4.90 × 10^−10^
RI	4.07 × 10^−9^	6.70 × 10^−9^	2.03 × 10^−8^	0.00	1.60 × 10^−7^	1.76 × 10^−9^	6.77 × 10^−9^	1.02 × 10^−8^	6.37 × 10^−9^	1.33 × 10^−7^

Note: The values larger than the threshold value (HQ, HI, 1.00; RI, 1.00 × 10^−6^) were marked in bold.

## Data Availability

The data presented in this study are available on request from the corresponding author.

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
