# Peer review of "Characteristics of PM2.5 in an Industrial City of Northern China: Mass Concentrations, Chemical Composition, Source Apportionment, and Health Risk Assessment"

_ijerph, 2022, doi:10.3390/ijerph19095443_

Round 1

Reviewer 1 Report

Manuscript id: ijerph-1635727-peer-review-v1

Titled: Health risk of trace elements in ambient PM2.5 in an industrial city of Northern China

Major Revision.

Some examples are given below:

  • Section 2.1. The detailed sampling method should be written in this manuscript.
  • Write the details about sampling locations, activities, major emission sources. If possible, put in the table including latitude and longitude. Section 2.1 and 2.2 should be written in detail.
  • Graphical representation of locations should be mentioned on the map. Method part should be written and detailed.
  • If possible, make a table for risk factor data.
  • In the results section, the discussion part is very weak. The authors only represented the value of the results but it should be discussed in detail with other studies. Why Please rewrite.
  • The entire paper should be thoroughly checked and discussed clearly.

Reviewer 2 Report

The manuscript (MS) by Bai et al. provided a simple report on PM2.5 and PM-bound elements concentration at an urban and suburban area in Zibo, China. This work is primarily descriptive and is basically reporting measurement data with insufficient analyses. With the format shown here, this work represented a project report and/or technical report rather than a scientific article since it lacked in-depth discussion relating to the titles and sub-titles included. The author(s) also didn’t fully explore the data presented in Tables and Figures, which made some of the arguments highly speculative. Besides, the method and discussion section need to be strengthened and the author(s) should consider including more data explanation to improve clarity. Language use in this MS could use more paraphrasing and better wording to improve the clarity for readers. Additional concerns are expressed further below:

Abstract:

Line 19-20: What conclusions could be drawn from these HI and TCR values?

Line 10: Better to use ‘each season’ with ‘every season’ or ‘four seasons’

Line 14-15: Replace ‘pollution sources and chemicals’ with ‘pollution sources’

Introduction:

Line 51-52: should be put in the Methodology section

Method: Although the authors cited their previous work to support the method section. I would suggest including some important information in the method section to clarify for the readers.

Line 57-58: More information regarding the technical details of the sampler should be provided.

Line 59: more information on the sampling sites’ coordinates should be provided

Line 64: ‘other elements’ should be specified

Line 69-70: information regarding CMB model should be briefly explained here. Particularly how the authors chose the potential sources (i.e. 9 elements attributed to 9 sources?)

Line 73-80: the sources’ information could be a little bit unclear. Are these sources defined by the program or by the author? If manually defined, such information should be listed in the Results & Discussions section.

Line 82: Health risk of PM2.5-bound heavy metals could be evaluated for dermatology and ingestion pathways also (USEPA, 2011). Has the author tried to evaluate non-carcinogenic and carcinogenic indices for these exposure pathways?

USEPA (2011) Exposure Factors Handbook 2011 Edition (2011). U.S. Environmental Protection Agency, Washington, DC, EPA/600/R-09/052F.

Line 93: Was body weight taken into account for the calculation? (reference 31)

Results and Discussions:

From my perspective, this MS hasn’t fully interpreted the dataset as well as drawn speculative discussion and/or conclusion due to the lack of reference manipulation. The author(s) need to provide on-point, relevant data and/or evidence to further clarify the argument. For example, comparison with other studies or inter-element statistical comparisons (i.e. Pearson correlation coefficients) should be made to better explore the dataset and draw out necessary conclusions. Besides, the way figures are presented throughout the MS should be further adjusted.

Line 108: Replace “Figures” with “Figure”

Line 110-115: Necessary comparison between the basic PM2.5 statistics of two sites should be included here. Furthermore, comparison with sites with the same characteristics or previous studies should be included to highlight the PM2.5 characters. Besides, relatively few details on temporal variation were described in this section.

Figure 1: The curve lines in these figures were not explained and include in the previous discussion.

Line 120-125: Information on 16 elements can be grouped into sub-groups for clearer comparison and better interpretation in the following sections

Line 133-134: This sentence could be removed.

Line 135-137: Conclusion should be included after this argument. In addition, these statements are speculative and have no supporting refs or data.

Line 139: Replace ‘improve’ with ‘increase’

Line 139-140: From lines 132-133, an additional explanation should be made before jumping to the conclusion in this sentence.

Line 141-145: The argument shown in these two sentences seemed detached from the above information. Comparison with regulation standards should be demonstrated at the beginning of the paragraph.

Figure 2: Adjust the y-axis.

Line 152: Are there any statistical evidence to choose a 9-source CMB solution for 24 elements? From Figure 3, it seems like elements with major abundance (i.e. Ca, Al, Fe) can be removed to achieve a clearer solution.

Line 156-158: Modify this statement.  

Line 158-160: Redundant information.

Line 161-163: These are speculative and unclear. How lower CD rates can explain better environmental policies?

Line 171-172: Readers should expect the measured metal concentration (i.e. analyzed concentration) as input for health risk assessment, not simulated data from CMB model.

Line 175: List the ‘acceptable range’ value.

Line 176-177: What do the comparison with values from a Taiwan study (Chen et al., 2013) and Liaoning study (Guo et al., 2018) imply?

Line 185: Except for TE, what about other indices? Do they pose a higher cancer risk too, and what implication can be drawn here? Similar analysis and/or comparison should be shown for lines 185-191

Line 192-199: Figures (i.e. values) should be included to further clarify this paragraph. Author(s) can compare the source contribution values from other studies with values obtained in this study, make necessary arguments before jumping to the conclusions regarding policy implementation.

Conclusion:

Line 203: It should be near 50% (or ‘one-half’) according to line 111 for a better conclusion.

Line 206: A conclusion should be included here based on the discussion of different Ni temporal variations in Section 3.1.

Line 210-212: Perspective figures for the contribution of traffic emission (20%?) should be included here. The term ‘elevated metal concentrations’ should be further explained.   

Reviewer 3 Report

Dear Author,

Please consider the following suggestions and comments.

General comments:

A) Given the special susceptibility of children, the health risk assessment should be performed for adults and for children (e.g. as in ref. 38), taking into account the differences in body weight and inhalation rate.

B) The data that is presented in Table S1, pertaining to the concentrations of ions and OC/EC, should be presented graphically in Fig. 2, along with the elements. The respective mean concentrations +-SD should be presented in the text, namely for total water-soluble inorganic ions and SIA. The NO3-/SO42- ratio should also be discussed.
Furthermore, a comparison with the results in ref. 22 should be performed.

C) Luo et al. (2018) (https://doi.org/10.1016/j.atmosres.2018.05.029) found that other sources are important for PM2.5 in Zibo, a petrochemical city, namely: petrochemical industry (contributing to 14.9 % of PM2.5) and biomass burning (contributing for 11.3% of PM2.5). The authors found that the vehicular emission factor accounted for only 2.8% of PM2.5. According to Luo (2018), the main contribution of aerosols in Zibo was soil dust, but in the present work the contribution was much lower. These major differences between the two studies, relative to Zibo, must be addressed in the discussion. 

D) It was difficult to find some references (e.g. 19-21) and some could not be found (ref. 24; 25, I could find only a preprint). Please add DOI to all references.

Specific comments:

Line 31: More general and recent references could be cited here, pertaining the health effects of PM2.5., e.g. Manisalidis et al., Environmental and Health Impacts of Air Pollution: A Review. Front. Public Heal. 2020, 8.

Line 43: 24-hour AQG level is 15 ug/m3 (not 25 ug/m3) according to the WHO Global Air Quality Guidelines 2021; and please insert reference.

Line 47: Could not find references 24 and 25 (only a preprint of the latter); please refer DOI in all references.

Line 48: It is not true that receptor modelling has not been applied in PM2.5 in Zibo in the literature: see Atmospheric Research 212 (2018) 285-295, where PMF model was applied by Luo et al.. This publication is crucial for comparison with the present work, but it was not cited, a gap that must be addressed in the whole discussion aspects.

Line 52: A reference for CMB model should be included here.

Line 54: A study area description, including a figure for geographical contextualization of the sampling sites, should be included.

Line 67: The analytical techniques should be briefly mentioned in this sub-section (although they are described in ref. 22).

Line 67:  "water-soluble ions" (instead of "water ions")

Line 69: The authors should address why Positive Matrix Factorization (PMF) model was not used in this work.

Line 70: Reference 27 describes the application of Chemical Mass Balance receptor model CMBv8.2, US-EPA, which is not the model used in the present paper (CMB CRAES1.0). Thus, the description of CMB CRAES1.0 is required, namely with the indication of the electronic reference where the model can be downloaded for public use, with the indication of other published works where the model has been applied and the indication of the main differences from EPA CMB.

Line 88: IRIS electronic refence should be added.

Line 91: Please add reference here, for the applied health risk assessment expressions.

Line 94 (eq. 3): This expression is not correct (remove typo).

Line 95: typo (";").

Line 95: μg.m-3 or μg/m3 can be used, but standardization should be performed throughout the document.

Line 95: remove "And" in the begining of the sentence.

Line 97: It must be specified if the median, the mean or the 95% upper confidence limit of the mean was considered in the concentrations (and why).

Line 101: ")" is missing.

Line 102: remove "." in the begining of the sentence.

Line 102: "f" is subscript. Should be "RfCi".
Line 102: Please specify the reference for RfCi values considered. 

Line 107: "spatial variation" is not a very adequate expression here, since only 2 sites are compared.

Line 111: typo (",")

Line 112: During the year, sampling certainly included non-haze and haze days and this aspect should be addressed in the paper.

Line 112: The Chinese National Ambient Air Quality Standard (GB 3095-2012) should be explicited and in parallel to this (high) limit, class 1 (35 ug/m3) and the WHO guidelines should also be taken into account in the discussion.

Line 112: The electronic reference should be included in ref. 33.

Line 114: remove "And" in the begining of the sentence.

Line 118: instead of "mean value", "daily mean value" or "24-hour mean value".

Line 119: add: ..."and also the central 90% data (5th -95th percentiles; whiskers)."

Line 122: put "respectively" at the end of the sentence.

Line 123: Put the medians in brackets after each element, e.g. Al (384.7), adding "(medians in ng/m3 inside backets)" at the end of the sentence.

Line 126: Put the medians in brackets after each element.

Line 129: "p" in lower case letter.

Line 128: In the Materials and Methods section, the statistical test (and statistical software) used must be specified (with a brief justification).

Line 130: The statistical significant differences for each element, between seasons and also between sites for the same season, should be indicated in Fig. 2, by the use of * or letters (a, b, ...) above each boxplot.

Line 135: I don't think ref 34 is correct.

Line 140: ..."Ni had a different source from the other elements."

Line 145: the reference is missing.

Line 151: Please explicit which species were used in the CMB modeling (which elements and/or ions, OC/EC?) and if some variable was discarded. 

Line 156: The main tracers of each source considered in CMB modeling must be specified, mentioning the reference.

Line 160: "percentage contribution", instead of "rate".

Line 162: Place a reference relative to the management policies undertaken in Zibo, if possible.

Line 166: remove "Respective"

Line 167: This specification of the meaning of each acronym makes more sense in the legend of Table 1 (the first time the acronyms are used).

Line 173: "total average risk of cancer", instead of "total risk of cancer" (since the average of the two sampling sites was determined).

Line 179: Reference 37 is incorrect (in ref. 37 health risks were not calculated). 

Table 2: formatting (1st row).

Lines 183 and 184: express all values in scientific notation.

Lines 184 and 187: "lower than 10-6", instead of "< 10-6".

Line 189: graphite production should also be a focus of attention.

Line 183: In this paragraph, the specific species that presented carcinogenic risk must be mentioned. The species that exceeded guideline limits published by WHO must also be mentioned.

Round 2

Reviewer 2 Report

Some improvements can be found in this revision of this MS but it still looks like a simple report on PM2.5 and PM-bound elements concentration. In this revised work, albeit included with a more detailed analysis, the overall structure was not thoroughly checked. The format of tables and figures should be adjusted, and proofreading is strongly suggested. Traffic emission was highlighted but the role of other sources was not portrayed with proper evidence (e.g. haze pollution). Additional data from EC/OC didn’t provide any important information regarding the major finding of this study. Besides, the arguments in Results & Discussion section still need to be strengthened to better support the data analysis and MS clarity. Therefore, my evaluation of the manuscript remains the same. Additional concerns are expressed further below:

Abstract:

Line 20-21: Information regarding Cr(VI) concentration should be discussed in Result & Discussion section.

Line 22-28: Combine to shorter sentences to highlight the proportion of vehicle emissions.

Line 30-32: Wearing masks is not a plausible solution to prevent PM2.5 pollution

Introduction:

Line 52-53: The authors might want to use a better classification method for the items in this sentence (i.e. only list out specific types of industry)

Line 53-56: This sentence is not suitable to put here.

Line 56: Should be ‘CMB’ model

Line 58-62: The objectives of this study should be explained more clearly.

Method:

Figure 1: Unfinished title. This figure also did not show clearly the description of the surrounding (i.e. residential, administrative, commercial buildings) shown in lines 67-71.

Line 92: The IMPROVE full name should be written with capitalized first letters.

Line 110-115: More information should be provided about the different sample preparation methods (acid vs. alkali) and different analysis methods (ICP-MS and ICP-OES)

Line 117-119: From reference [31], data number was not a suitable choice for preference of CMB model over PMF.

Line 138: percentage of explained mass should be < 100% (?)

Line 172-175: Information provided in this section is redundant and thus could be removed.

Results and Discussions:

From my perspective, the authors tried to include more data analysis for a better presentation, but they did not support each other well as a whole. The author(s) need to provide on-point, relevant data and/or evidence to further clarify the argument. For example, little information regarding the roles of haze pollution and traffic emission were discussed, although it was mentioned both in Abstract & Conclusion sections. The chosen elements for detailed discussion in correlation analysis should complement or support the sources profile in CMB model, or vice versa.

Line 188-189:  How could higher PM2.5 concentration reflect more severe haze pollution?

Line 200: The speculation of ‘normal curve’ is still not clear.  This part could be removed for better data presentation.

Line 206: The term SIAs was not introduced previously.

Line 224-225: Please include OC & EC relative proportion in PM2.5 to demonstrate this point.

Line 241-244: Please provide references or supporting information to show the link between decreasing OC concentration and the shift in energy use.

Line 256-257: Listed number should not be repeated.

Figure 7: Further explanation regarding the unit and chart components should be provided. Should elements with near-zero contribution (e.g. Pb, Ba, Cd, Sr, As…) be removed to obtain better simulation result?

Line 299: So the source profiles here were subjectively chosen?

Line 315-316: Can EC correlations with other PM2.5 components be shown to emphasize its indicating feature for traffic emission?

Conclusion:

Line 365: The contribution was not clarified clearly during the CMB and correlation analysis result.

Line 371-374: What does the high Cr concentration that exceeded the AQ standard imply?

Line 382-383: You mean for other metals? Otherwise, this sentence contradicts the previous sentence.

Reviewer 3 Report

Dear Authors

Please consider the following comments on your revised version of the paper:

-The inexistence of a petrochemical source profile, contrary to Luo et al. (2018) (https://doi.org/10.1016/j.atmosres.2018.05.029), should be better addressed. Is there petrochemical industry at Zibo nowadays?

-Line 130: It is stated that: "The source profiles were conducted based on previous field research." Please specify reference, if possible.

Line 298: It is stated that: "Source profiles... have been prepared for Zibo."

It is still not clear how the source profiles have been prepared. Please specify further, namely add reference for source profile for graphite and glass production.

Refs 48 and 49 are very old. Please replace by more recent references.

The paper should be thoroughly checked for typos and spelling errors which still exist.

Best regards
